# Effect of Exclusive Enteral Nutrition and Corticosteroid Induction Therapy on the Gut Microbiota of Pediatric Patients with Inflammatory Bowel Disease

**DOI:** 10.3390/nu12061691

**Published:** 2020-06-05

**Authors:** Lara Hart, Yasamin Farbod, Jake C. Szamosi, Mai Yamamoto, Philip Britz-McKibbin, Camilla Halgren, Mary Zachos, Nikhil Pai

**Affiliations:** 1Department of Pediatrics, Division of Gastroenterology, Hepatology and Nutrition, McMaster Children’s Hospital, McMaster University, Hamilton, ON L8N 3Z5, Canada; lara.hart@medportal.ca (L.H.); yasaminfar71@gmail.com (Y.F.); camilla.halgren@medportal.ca (C.H.); zachosm@mcmaster.ca (M.Z.); 2Farncombe Family Digestive Health Research Institute, Department of Medicine, McMaster University, Hamilton, ON L8N 3Z5, Canada; jc.szamosi@alumni.utoronto.ca; 3Department of Chemistry and Chemical Biology, McMaster University, Hamilton, ON L8N 3Z5, Canada; yamamm@mcmaster.ca (M.Y.); britz@mcmaster.ca (P.B.-M.)

**Keywords:** microbiome, exclusive enteral nutrition, pediatric inflammatory bowel disease, induction of remission

## Abstract

Introduction: Exclusive enteral nutrition (EEN) and corticosteroids (CS) are effective induction therapies for pediatric Crohn’s Disease (CD). CS are also therapy for ulcerative colitis (UC). Host–microbe interactions may be able to explain the effectiveness of these treatments. This is the first prospective study to longitudinally characterize compositional changes in the bacterial community structure of pediatric UC and CD patients receiving EEN or CS induction therapy. Methods: Patients with diagnoses of CD or UC were recruited from McMaster Children’s Hospital (Hamilton, Canada). Fecal samples were collected from participants aged 5–18 years old undergoing 8 weeks of induction therapy with EEN or CS. Fecal samples were submitted for 16S rRNA sequencing. The Shannon diversity index and the relative abundance of specific bacterial taxa were compared using a linear mixed model. Results: The clustering of microbiota was the highest between patients who achieved remission compared to patients still showing active disease (*p* = 0.029); this effect was independent of the diagnosis or treatment type. All patients showed a significant increase in Shannon diversity over the 8 weeks of treatment. By week 2, a significant difference was seen in Shannon diversity between patients who would go on to achieve remission and those who would not. Conclusion: The gut microbiota of pediatric UC and CD patients was most influenced by patients’ success or failure to achieve remission and was largely independent of the choice of treatment or disease type. Significant differences in Shannon diversity indices occurred as early as week 2 between patients who went on to achieve remission and those who continued to have active disease.

## 1. Introduction

Pediatric inflammatory bowel disease (IBD) comprises 25% of incident cases of IBD [1]. Exclusive enteral nutrition (EEN) and corticosteroids (CS) are both effective treatment options for the induction of remission in pediatric Crohn’s disease (CD), with a clinical remission rate of approximately 80% for both treatments [2,3]. In the management of pediatric IBD, EEN is recommended as a first-line treatment for pediatric CD. EEN leads to significantly higher rates of mucosal healing and avoids short- and long-term corticosteroid side effects that disproportionately affect children [4]. EEN is also beneficial for nutritional rehabilitation, catch-up growth, and improved bone density [4,5,6,7]. There is a paucity of literature on the role of EEN in ulcerative colitis (UC) [8].

The mechanisms underlying the effectiveness of EEN in pediatric IBD remain unclear. A microbiome-mediated pathogenesis has been postulated and continues to be studied. Two studies have shown that EEN produces profound changes in a microbial community’s structure within as little as 24 h. In particular, *Bacteroides* spp. and *Prevotella* spp. significantly decreased, with significant increases in *Clostridium leptum* abundance following EEN therapy [5,9]. These changes correlated with improved disease activity scores. Acute colitis in IBD also carries distinct microbial signatures. Patients with active UC have been shown to have lower *Bacteroides* spp. and *Clostridium* spp. [10,11]. Conversely, active CD shows increased *Enterobacteriaceae* spp. and decreased *Clostridium* spp. [5]. 

The role of EEN therapy in UC remains unclear. One study showed improvements in the disease activity and markers of systemic inflammation but no head-to-head comparisons against CS have been performed to date [12]. Clinical guidelines do not describe a role for EEN in the treatment of acute UC [13]. Baseline differences in the taxonomy between IBD phenotypes may account for differences in EEN’s efficacy between CD and UC. Additionally, few studies have described the effects of CS for the induction of remission in UC or CD on bacterial community structure, or longitudinal changes in the microbiome of patients on CS or EEN therapy for the induction of remission in pediatric IBD [5,14,15,16]. The purpose of this study was to report the serial clinical, biochemical, and microbiome changes that occurred in children using EEN versus CS for the induction of remission in pediatric IBD.

## 2. Materials and Methods 

### 2.1. Study Design and Setting

We conducted a prospective cohort study on patients admitted to the inpatient Pediatric Gastroenterology service at McMaster Children’s Hospital, Hamilton, Ontario (Canada), with a diagnosis of IBD (CD, UC) over a 9-month period. Participants were recruited from September 2015–June 2016. The study received approval from the Hamilton Integrated Research Ethics Board (#15-365). 

### 2.2. Participants

Patients that were 5–18 years old with IBD (CD or UC), diagnosed through histopathology from endoscopic tissue biopsies, serum bloodwork (complete blood count, albumin, sedimentation rate, C-reactive protein (CRP)), fecal calprotectin (FC), and small bowel imaging (magnetic resonance enterography) were approached for recruitment. Patients met the inclusion criteria if they were being initiated on EEN therapy or intravenous CS for the induction of remission of CD or UC. The choice of treatment was at the discretion of the on-service, attending pediatric gastroenterologist. Patients were excluded if they were younger than 5 years old, received antibiotic therapy, or did not require hospital admission for the initiation of treatment. Patients were also excluded if they were started on a 5-ASA (aminosalicylic acid) or biologic therapy for the induction of remission. Two trained research assistants assessed eligibility, obtained informed consent, and addressed all communication requests from patients and families. Patients were approached consecutively over the 9-month study period. 

### 2.3. Blinding

This was a non-blinded study. The decision to treat with CS or EEN was at the discretion of the on-service, attending pediatric gastroenterologist.

### 2.4. Intervention

All patients received 8 weeks of EEN or CS treatment. EEN therapy was initiated in the hospital and provided via a nasogastric tube. Patients received Peptamen^®^ 1.5 at 120% of their daily caloric needs (based on ideal body weight calculations). Patients were discharged home once the full volume of EEN feeds was reached and tolerated (48–72 h after initiation). Feeds were condensed to 12–14 h over the first week. Volumes could be increased if the patient was hungry and the rate could be decreased if there were side effects, such as discomfort or distension. Other than formula, patients were allowed to consume only clear fluids orally. An unrestricted diet was reintroduced after 8 weeks in accordance with our hospital-specific EEN realimentation weaning protocol. Patients on CS therapy were started on treatment in the hospital with IV methylprednisolone at 1 mg/kg/day (max: 40 mg/day). Once symptoms improved, patients were transitioned to oral CS and discharged home on 1 mg/kg/day PO (per os) x 2 weeks, followed by a progressive wean by 5 mg per week. Additional maintenance therapies were initiated as early as two weeks into either EEN or CS induction therapy by the treating pediatric gastroenterologist.

### 2.5. Sample Collection

Fecal samples were collected for microbial analysis before initiating induction therapy and sequentially during induction therapy (days: 0 (pre-treatment), 3, 7; weeks: 2, 4, 6, 8, 12). The samples were immediately stored in a 4 °C refrigerator and samples brought from home were placed on ice until they could be re-refrigerated in the gastroenterology clinic. Fecal samples were then aliquoted for storage at −80 °C within 4–8 h of collection before sequencing. Serum bloodwork (complete blood count, albumin, CRP) and FC were collected at similar time points to the fecal microbiome sample collection times (Figure 1). Detailed instructions were provided to families and study team members to ensure fecal samples were handled and processed similarly, including between inpatient and outpatient collections.

### 2.6. Clinical Assessment

Once weekly, while on induction therapy, patients received a telephone call from trained research assistants to assess clinical symptoms. Clinical assessments were measured using the Pediatric UC Activity Index (PUCAI) or a modified Pediatric CD Activity Index (PCDAI) (excluding the physical exam information) [17,18]. Clinical improvement was defined as a decrease in the PUCAI or modified PCDAI from baseline enrolment, and clinical remission was defined as a score of ≤10 for each scale. The response to treatment was classified in terms of active disease, clinical response, clinical remission, or full remission. Active disease was defined as all baseline clinical and biochemical (CRP, albumin, hemoglobin, FC) parameters. Clinical response was defined as improvement in PCDAI/PUCAI and biochemical parameters from active disease. Clinical remission was defined as clinical response and improvement in biochemical markers from active disease. Full remission was defined as clinical remission and biochemical markers within normal limits (FC < 250 mcg/g).

### 2.7. Microbial Community Structure Analysis

Microbial community profiling was conducted by extracting genomic DNA from patient and donor stool samples using a protocol described previously, which enhances total DNA recovery [19]. After genomic DNA extraction, the V3 region of the 16S ribosomal RNA gene was amplified (total polymerase chain reaction volume of 50 μL (25 pmol of each primer, 200 μL each deoxynucleoside triphosphate, 1.5 mM MgCl_2_, and Taq polymerase), divided into triplicate for greater efficiency). Samples were sequenced using the Illumina MiSeq platform (San Diego, CA, USA), as per manufacturer’s instructions and raw sequence reads were processed as described previously [20]. The stool microbiota was sampled to a mean depth of 103,341 reads per sample (median = 87,509). 

### 2.8. Statistical Analysis

For clinical outcomes, statistical differences between pairwise comparisons were calculated using the Mann–Whitney U test, with significance defined as *p* < 0.05. Microbiome data were analyzed in R using the Phyloseq package (version 3.11; Boston, MA, USA). Samples were ordinated using a principal coordinate analysis (PCoA) of Bray–Curtis distances to visualize clustering. The differential abundance of bacterial genera was tested using the generalized linear model implemented in DESeq2 (version 3.11; Boston, MA, USA) and restricted to those genera that were present in at least 50% of the samples. We omitted any mean <10 reads/100,000 (relab <0.001), max <100 reads/100,000 (relab < 0.1). If *p*(0) ≤ 0.7, we performed DESeq directly, and if *p*(0) > 0.7, we binarized and performed logistic regression on the results. Spearman correlations of the genus relative abundance with hemoglobin, albumin, CRP, and FC were also restricted to those genera present in at least 50% of the samples using pairwise-complete observations.

## 3. Results

### 3.1. Participants

Thirty patients were enrolled (69% male); 16 patients received EEN and 14 received CS therapy. Of these, all patients receiving EEN had CD and 29% of patients receiving CS had CD. A total of 63% of patients were newly diagnosed and undergoing their first induction therapy. Baseline bloodwork and FC values were similar in patients who received EEN and CS (Table 1).

### 3.2. Response to EEN or CS

A total of 60% (*n* = 6) of patients with UC and 90% (*n* = 18) of patients with CD achieved clinical remission by the end of therapy at week 8. Of the 6 patients who did not achieve remission, 83% (*n* = 5; 1 CD, 4 UC) received CS and 17% (*n* = 1; CD) received EEN. One patient required a second hospitalization for disease relapse (despite CS therapy). For both the EEN and CS treatment groups, there was a significant decrease in CRP (EEN, CS: *p* = 0.03), increase in albumin (EEN, CS: *p* = 0.02), and decrease in FC (EEN: *p* = 0.033, CS: *p* < 0.005) after therapy (Table 2).

### 3.3. Microbiota Composition in EEN vs. CS Treated Patients

When all time points were included, the microbiota of patients who had CD or UC clustered differently based on principal coordinate analysis (Figure 2) and a PERMANOVA test (R^2^ = 0.047, *p* = 0.001). Bacterial taxon abundances were measured over the course of treatment between patients who received CS (*n* = 14; 10 UC, 4 CD) and EEN therapy (*n* = 16; 0 UC, 16 CD). Overall, as patients progressed from active disease to full remission, greater abundances of the following genera were found in both treatment groups: *Blautia* (*p* = 0.045), *Sellimonas* (*p* = 0.002), and uncharacterized bacteria from family Ruminococcaceae (*p* = 0.0032) (Figure 3A). The abundances of three genera were reduced as the patients’ health improved: *Granulicatella* (*p* = 0.027), *Haemophilus* (*p* = 0.0001), and *Streptococcus* (*p* = 0.002) (Figure 3B). In two taxa, there was a significant association of the disease state with abundance but no clear linear pattern of change: *Bacteroides* (*p* = 0.03) and uncharacterized bacteria from family Peptostreptococcaceae (*p* = 0.00005) (Figure 3C). As patients progressed from active disease to full remission, while not meeting statistical significance, greater abundances were also found in both treatment groups for *Faecalibacterium* and *Fusobacterium* that were absent from more than 50% of the samples in the remission state. These changes were seen regardless of treatment type at the genus and phylum levels (Figure 3D,E). 

At the end of therapy (week 8), although not reaching statistical significance, patients treated with EEN (*n* = 16; 16 CD) showed depletion in bacteria from the *Fusobacterium*, *Escherichia*/*Shigella*, and *Veillonella* genera. In patients treated with CS (*n* = 14; 10 UC, 4 CD), reductions in *Alistipes*, *Veillonella*, and *Fusobacterium* genera were observed. These changes were independent of the patients’ underlying disease type (CD, UC) (Figure 4).

The change in microbiota composition was also plotted based on clinical and serological markers of response to treatment (active disease, clinical response, clinical remission, full remission). A permutation test of the microbial community dispersions during active disease between UC and CD showed significant differences between the two diagnoses during active disease (F(3) = 7.1094, *p* = 0.011). Tighter clustering was observed in patients after 8 weeks of therapy, independent of treatment type, while the greatest dispersion was seen in patients during active disease before treatment was initiated (Figure 5A,B).

We also noted a significant increase in Shannon diversity over time (F(1) = 8.019, *p* = 0.006) and across the disease states (F(3) = 4.389, *p* = 0.007). This effect did not vary based on the type of treatment received (Figure 6A,B). We also found a significant difference in Shannon diversity at week 2 of treatment between patients who would go on to achieve remission at end of therapy (week 8) and those who would not. This was found in all patients, regardless of whether they received CS or EEN (F(1) = 5.074, *p* = 0.044) (Figure 6C), and was not present during the active disease state at baseline (week 0).

### 3.4. Correlations between Bacterial Genera and Clinical, Fecal, and Serum Markers of Disease Status

The genera *Sellimonas, Ruminiclostridium,* and families Ruminococcaceae and Erysipelotrichaceae were strongly positively correlated with serum hemoglobin and albumin levels, while *Veilonella*, *Escherichia*/*Shigella,* and *Haemophilus* were strongly negatively correlated with both serum levels. CRP and FC were negatively correlated with *Sellimonas, Ruminiclostridium,* and families Ruminococcaceae and Erysipelotrichaceae (Figure 7).

## 4. Discussion

This prospective cohort study in pediatric UC and CD was designed to assess dynamic changes in the intestinal microbiota profiles of patients receiving eight weeks of induction therapy using CS or EEN. Our study found that regardless of disease phenotype or therapy modality, at the end of 8 weeks, as patients’ health improved, their microbiome communities were significantly more closely clustered than they were before treatment began. 

Both treatment options achieved the induction of remission in the majority of UC and CD patients. Following treatment with EEN, the median decreases in CRP and FC were 19.6 mg/L and 2805 mcg/g, respectively; PCDAI scores decreased by 19.0 points. Patients receiving treatment with CS had a median decreases in CRP and FC of 21.1 mg/L and 2523 mcg/g, respectively; there was a decrease in PCDAI of 23.0 points in CD patients and a decrease in PUCAI of 37.0 points in UC patients. These results were consistent with previous work demonstrating a response to CS and EEN treatments in UC and CD [2,4,8]. 

During the treatment phase in our study, a significant shift was demonstrated in the abundances of some microbial taxa in patients from both UC and CD groups. Patients who were treated with EEN showed a marked depletion in bacteria from the *Fusobacterium*, *Escherichia/Shigella*, and *Veillonella* genera. This is consistent with previous literature suggesting that the treatment response in CD is associated with changes in these taxa [14]. In patients treated with CS (*n* = 14; 10 UC, 4 CD), reductions in the *Alistipes*, *Veillonella*, and *Fusobacterium* genera were observed. Due to sample size limitations, we did not independently assess taxonomic differences in CS patients with UC and CD separately. Further, patients who achieved a state of remission had an increased abundance of *Blautia*, *Sellimonas*, and the family Ruminococcaceae, as well as a decrease in the abundance of *Granulicatella, Haemophilus*, and *Streptococcus* genera. These results suggest that if therapy is successful in inducing remission, the microbial community shifts toward a similar healthy end-state (remission microbiota profile), regardless of the patient’s underlying disease type or induction therapy. This could also suggest that at the mucosal level, UC and CD lie along a continuum of a common disruptive host–microbe pathway rather than showing two fundamentally distinct microbial disease states. This has previously been described by other investigators through genome-wide association studies, suggesting that shared genes coding for immune loci are found across both CD and UC patients [21,22,23].

We found that as patients entered remission, they showed a significant increase in Shannon diversity compared to during the active disease state. This occurred in patients treated with CS or EEN, which has been supported by both Schwerd et al. [24] and D’Argenio et al. [25]. This has been challenged by other investigators, where alpha-diversity was actually found to decrease in patients treated with EEN. We propose that our conclusions must be evaluated in consideration of our sample size, having collected 45 stool samples throughout the study among 16 patients who received EEN. Yet, other investigators have also drawn conclusions from similarly small sample sizes. Gerasimidis et al. (20 patients; *n* = 68 samples), Leach et al. (6 patients; actual number not reported, *n* = 48 samples maximum), and Kaakoush et al. (5 patients; actual number not reported, presumed *n* = 25–45 samples) all reported a significant decrease in alpha-diversity over time among pediatric patients receiving EEN [5,9,26]. All three groups reported on similarly limited sample sizes and stool samples for analysis. 

Unique to our study, we also noted a significant predictive effect of Shannon diversity as early as week 2 of treatment. Patients who went on to achieve remission by the end of therapy (week 8) had a significantly higher Shannon diversity in both treatment groups (CS, EEN) at week 2 compared to those who did not achieve remission. If reproducible, this finding would have significant implications for predicting responders from non-responders early in the treatment course. This would have appreciable impacts on the quality of life given that at week 2, patients generally have between 6–10 more weeks of treatment remaining. It could also provide a further biomarker of response to therapy, in conjunction with clinical findings and bloodwork investigations. 

Several bacterial taxa were associated with changes in clinical, serum, and fecal markers of disease status. *Ruminococcus* was positively associated with improvements in hemoglobin levels. This was consistent with other investigators who have shown similar correlations between *Ruminococcus* and improvement in hemoglobin in recipients of EEN [14]. For the first time, we report that reductions in CRP and FC were inversely correlated with changes in Ruminococcaceae.

There is little data available on the role of EEN in UC treatment. Animal models of dextran sulfate sodium (DSS)-induced colitis have shown that EEN can reduce cytokine expression and improve tissue histology [27]. A small case series from Wedrychowicz et al. studied the results of 39 pediatric UC and CD patients receiving EEN for induction therapy. This study did not compare results against corticosteroid induction but improvements in CRP, Hgb, clinical disease activity, and weight-for-age *z* scores were noted [12]. 

Our study has several limitations, particularly the small number of patients included and samples obtained, and therefore there was a treatment heterogeneity across CD and UC diagnoses. Our numbers reflect the difficulties in performing such trials in children and adolescents with IBD, where validated guidelines promoting the use of CS as first-line therapy in UC exist [8]. In our study, no UC patients received EEN as induction therapy. While our study protocol allowed primary pediatric gastroenterologists to assign their choice of induction treatment without restriction, patient and physician discomfort of assigning an atypical first-line therapy for acute UC limited our ability to compare the clinical and microbiota changes of this treatment approach. This precluded our ability to assess the role of EEN in patients with UC. Further, our institution heavily supports and encourages the use of EEN as first-line therapy in CD patients in accordance with international guidelines on the treatment of pediatric CD [2]. These limited the number of patients in the CD group who received CS for induction therapy but do provide some preliminary data that may be followed up on in future studies of a larger sample size.

## 5. Conclusions

In summary, our results suggest that UC and CD patients developed remission microbiota profiles that were similar. This trend was independent of the induction treatment choice (EEN vs. CS) and baseline diagnosis. While EEN therapy continues to be an important treatment option for CD, and CS remains the first-line treatment for patients with UC, broad changes in microbiota appear to be driven by the success or failure to achieve clinical remission more than treatment choice or the diagnosis itself. Further, a higher Shannon diversity at week 2 may predict the likelihood of remission at the end of therapy. Current work is underway to characterize dynamic changes in the fecal and urine metabolome of pediatric IBD patients to better delineate treatment responses to therapy.

## Figures and Tables

**Figure 1 nutrients-12-01691-f001:**
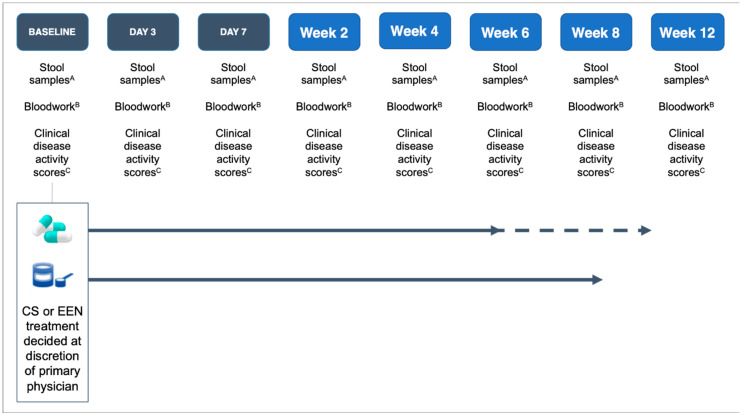
Overview of the study design. CS—corticosteroids, EEN—exclusive enteral nutrition. Note: CS therapy continued for 6–9 weeks at weaning doses; EEN therapy continued for 8 weeks. ^A^ Stool samples: fecal calprotectin and fecal microbiota. ^B^ Bloodwork: complete blood count (CBC), albumin, and C-reactive protectin (CRP). ^C^ Clinical disease activity scores: Pediatric Ulcerative Colitis Activity Index (PUCAI) and Pediatric Crohn’s Disease Activity Index (PCDAI).

**Figure 2 nutrients-12-01691-f002:**
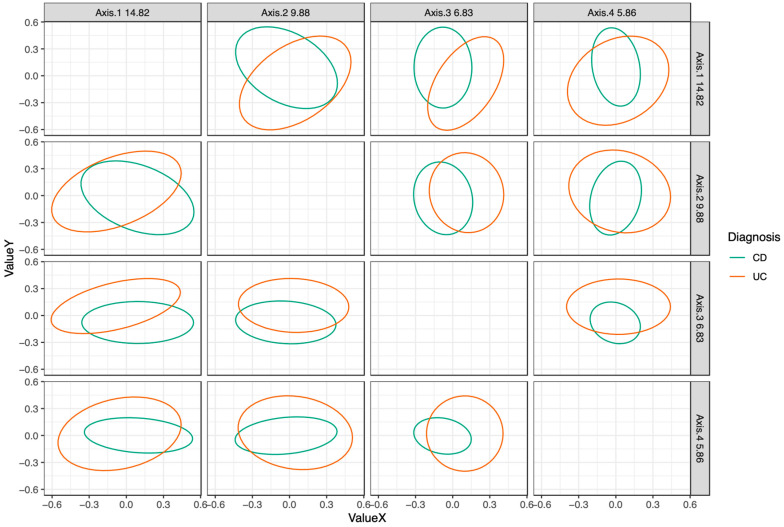
Principal coordinate analysis of CD and UC fecal microbiota at baseline (active disease). Fecal microbiota collected from 20 patients with CD, 10 patients with UC with active disease before the onset of treatment. Bacterial taxa clustered differently between diagnoses. CD—Crohn’s disease, UC—ulcerative colitis.

**Figure 3 nutrients-12-01691-f003:**
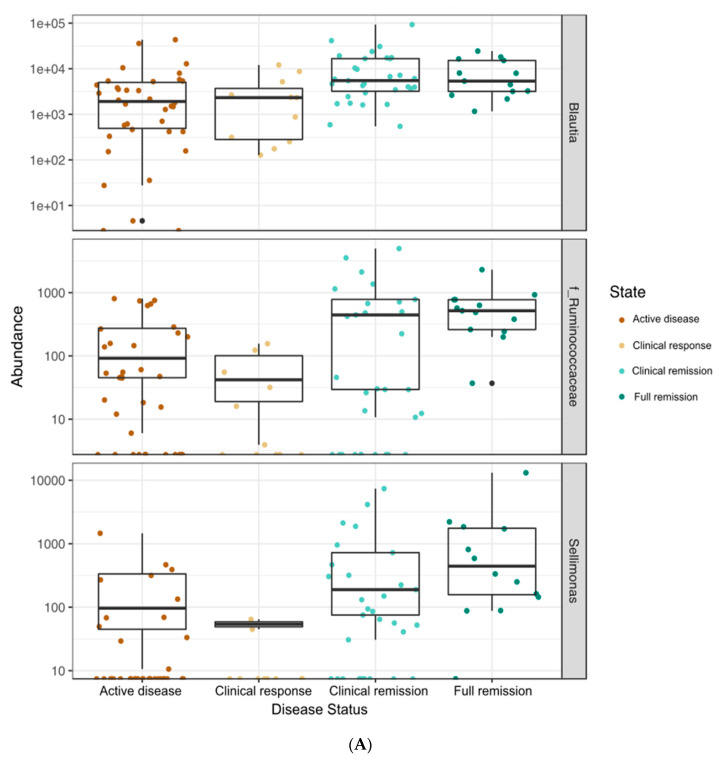
(**A**) Taxa that increased in abundance with disease response. Taxa that increased abundance over the course of treatment (as symptoms improved) were *Blautia* (*p* = 0.045), *Sellimonas* (*p* = 0.002), and uncharacterized bacteria from family Ruminococcaceae (*p* = 0.0032). (**B**) Taxa that decreased in abundance with disease response. Taxa that decreased in abundance over the course of treatment (as symptoms improved) were *Granulicatella* (*p* = 0.027), *Haemophilus* (*p* = 0.0001), and *Streptococcus* (*p* = 0.002). (**C**) Taxa with significant changes in abundance but an unclear disease response. There was a significant association of disease state with abundance but no clear linear pattern of change with *Bacteroides* (*p* = 0.03) and uncharacterized bacteria from family Peptostreptococcaceae (*p* = 0.00005). (**D**) The relative abundance of bacterial genera in terms of the treatment and disease state. The microbial community structure at the genus level of CD and UC patients receiving either CS or EEN therapy in terms of the phase of treatment. Greater abundances of *Bacteroides*, *Blautia*, and *Faecalibacterium* were noted from an active disease at the clinical response, clinical remission, and full remission stages in both treatment groups. (**E**) The relative abundance of bacterial phylum in terms of the treatment and disease state. The microbial community structure at the phylum level of CD and UC patients receiving either CS or EEN therapy in terms of the phase of treatment. CS—corticosteroids, EEN—exclusive enteral nutrition.

**Figure 4 nutrients-12-01691-f004:**
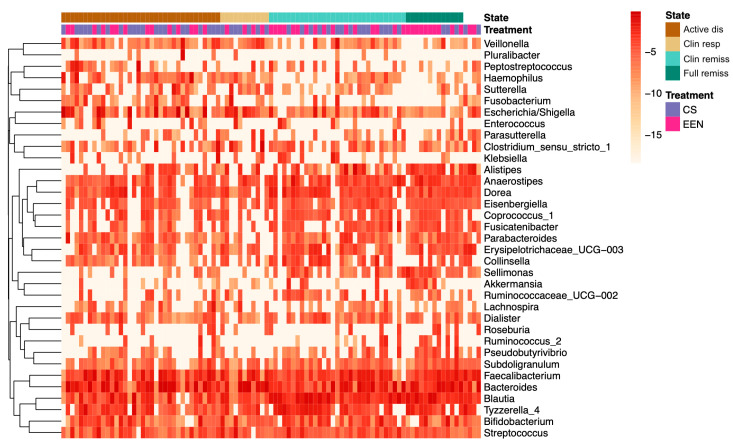
Heatmap illustrating the change in the taxonomic composition of the fecal microbiome in terms of the disease state and treatment type. Changes in the relative abundance of genera across the treatment state. At the clinical response stage, patients treated with EEN showed depletion in *Fusobacterium*, *Escherichia*/*Shigella*, and *Veillonella*, compared to patients treated with CS, who showed reductions in *Alistipes*, *Veillonella*, and *Fusobacterium* genera. CS—corticosteroids, EEN—exclusive enteral nutrition.

**Figure 5 nutrients-12-01691-f005:**
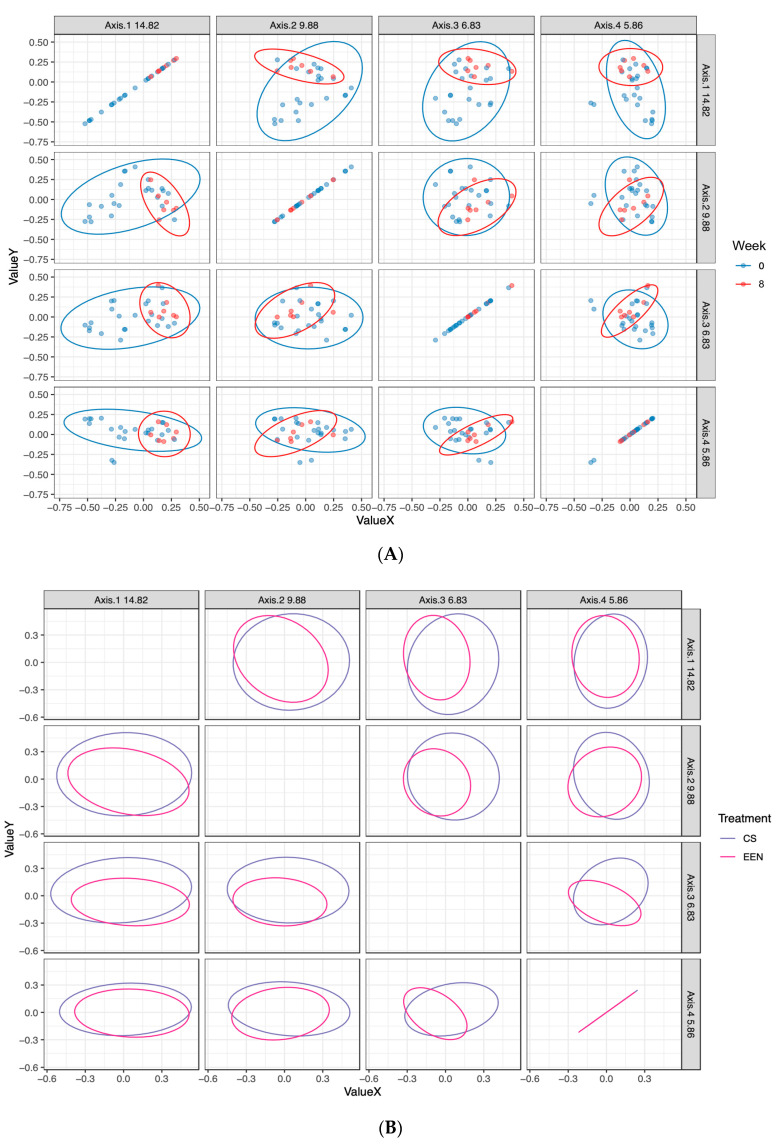
(**A**) Principal coordinate analysis of fecal microbiota at baseline (active disease) and end of treatment. Greater clustering was observed at the end of treatment independent of the disease type or treatment. (**B**) Principal coordinate analysis of the fecal microbiota in terms of the treatment type at the end of treatment. Principal coordinate analysis of patients at the end of treatment (week 8) in terms of the CS or EEN treatment type. Clustering was observed between all samples, independent of the treatment type. CS—corticosteroids, EEN—exclusive enteral nutrition.

**Figure 6 nutrients-12-01691-f006:**
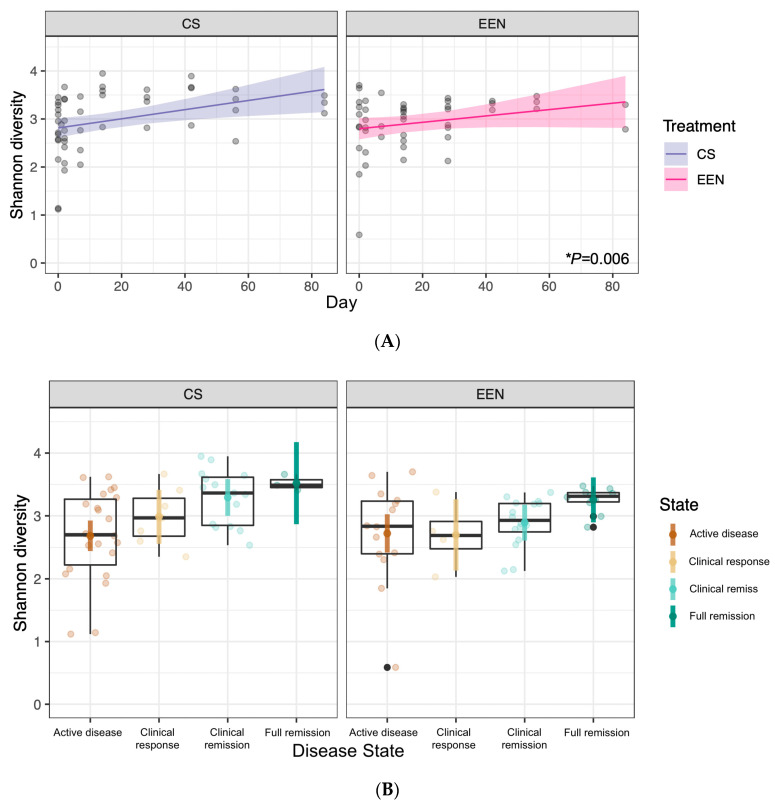
(**A**) Shannon diversity over time separated by treatment. A significant increase in Shannon diversity was observed over time (day 0 to day 84), independent of the treatment received (*p* = 0.006). (**B**) Shannon diversity over the disease state separated by treatment. A significant increase in Shannon diversity was observed over the disease state (active disease, clinical response, clinical remission, full remission) in both CS- and EEN-treated patients (*p* = 0.007). (**C**) Shannon diversity at week 2 compared to the final remission status at the end of treatment, separated by treatment type. A significant difference in Shannon diversity at week 2 was observed between patients who achieved remission at the end of treatment and those who did not achieve remission (*p* = 0.044). CS—corticosteroids, EEN—exclusive enteral nutrition.

**Figure 7 nutrients-12-01691-f007:**
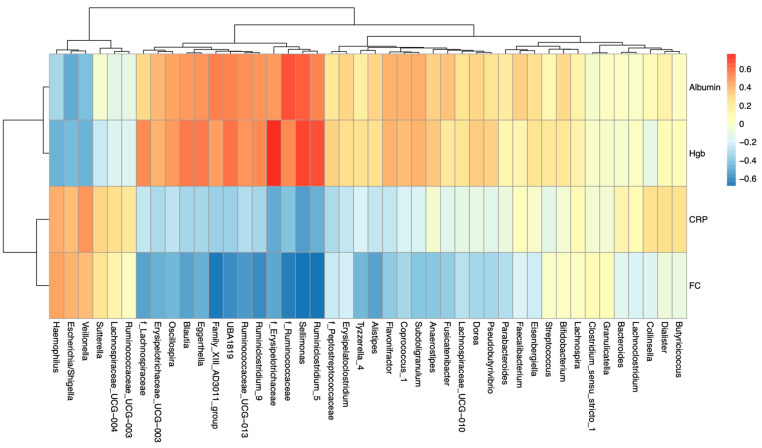
Correlation of fecal and serum markers of disease with bacterial genera. Genera *Sellimonas* and *Ruminiclostridium* positively correlated with serum Hgb and albumin levels, and negatively correlated with CRP and FC. Genera *Veilonella* and *Escherichia*/*Shigella* negatively correlated with serum Hgb and albumin levels. Hgb—hemoglobin, CRP—C-reactive protein, FC—fecal calprotectin

**Table 1 nutrients-12-01691-t001:** Baseline characteristics of patients starting induction therapy with EEN or CS.

Variable	Category	EEN (*n* = 16)	CS (*n* = 14)	*p*	TOTAL (*n* = 30)
Gender, *n* (%)	Male	11 (68.8)	7 (50.0)	0.80	18
	Female	5 (31.3)	7 (50.0)		12
Age, *n* (%)	≤12 years	5 (31.3)	7 (50.0)	0.80	12
	≥13 years	11 (68.8)	7 (50.0)		18
Diagnosis, *n* (%)	UC	0	10 (71.4)	0.94	10
	CD	16 (100)	4 (28.6)		20
Disease Activity Score, mean ± SD	PUCAI	--	54.5 ± 13.8	NA	10
	PCDAI	21.9 ± 10.8	31.25 ± 10.3	0.135	20
Stage, *n* (%)	New diagnosis	10 (62.5)	8 (57.1)	0.50	18
	Established disease	6 (37.5)	6 (37.5)		12
EEN Episode, *n* (%)	First-time EEN	13 (81.3)	--		--
	Repeat EEN	3 (18.8)	--		--
Investigations, median (IQR)	CRP (mg/L)	21 (7.8–46.2)	21.75 (10.3–35.8)	0.83	21.75
	Hgb (g/dL)	108.5 (97.5–188.8)	103 (93.0–117.0)	0.54	108.6
	Alb (g/L)	28 (25.2–32.2)	30.5 (28.2–33.5)	0.26	30
	FC (mcg/g)	3079 (1104–4305)	3069 (1105–4307)	0.95	2803

EEN—exclusive enteral nutrition; CS—corticosteroids; Alb—albumin; CRP—C-reactive protein; FC—fecal calprotectin; Hgb—hemoglobin; PCDAI—Pediatric Crohn’s Disease Activity Index; PUCAI—Pediatric Ulcerative Colitis Activity Index.

**Table 2 nutrients-12-01691-t002:** End of treatment characteristics of patients receiving EEN or CS.

Variable	Category	EEN (*n* = 16)	CS (*n* = 14)	*p*
Disease Activity Score, mean ± SD	PUCAI	--	17.5 ± 27.5	--
	PCDAI	2.5 ± 3.8	8.3 ± 10.4	0.18
Investigations, median (IQR)	CRP (mg/L)	1.45 (0.3–14.8)	0.7 (0–5.7)	0.67
	Hgb (g/dL)	119.5 (115.8–130.0)	113 (100.3–128.0)	0.39
	Alb (g/L)	36 (31.5–39.0)	35.5 (33.5–40.5)	0.78
	FC (mcg/g)	274 (104.7–1010.0)	546 (110.6–1093.0)	0.91

EEN—exclusive enteral nutrition; CS—corticosteroids; Alb—albumin; CRP—C-reactive protein; FC—fecal calprotectin; Hgb—hemoglobin; PCDAI—Pediatric Crohn’s Disease Activity Index; PUCAI—Pediatric Ulcerative Colitis Activity Index.

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
