# Peer review of "Effect of Exclusive Enteral Nutrition and Corticosteroid Induction Therapy on the Gut Microbiota of Pediatric Patients with Inflammatory Bowel Disease"

_nutrients, 2020, doi:10.3390/nu12061691_

Round 1
Reviewer 1 Report
Thank you for this interesting and well written article. You recognise the limitations of the study including sample size and the lack of CD patients in the CS group, limiting some comparisons.
I don't have any improvements to suggest
Author Response
Thank you for your review.
Reviewer 2 Report
This prospective cohort study reports on changes in faecal microbiota during treatment, with corticosteroids or exclusive enteral nutrition, of children with active inflammatory bowel disease. The major finding, that bacterial diversity increased similarly with treatment response in both groups, is interesting, and highlights that the direction of causality is unknown and the mechanism of action of EEN is unclear.
The study concept and design appear simple and well-executed. The use of language is clear and concise. Adequate review of the literature has occurred, and no ethical issues are evident.
This paper is a welcome addition to the literature.
The small study size is correctly identified as a major limitation – in particular, subgroup analysis is unreliable when some categories are so small (eg. 1 patient with Crohn’s disease did not achieve remission).
I think that the non-standard terms Response 1 and Response 2 would be better given descriptive names – eg. “clinical response” and “clinical remission”.
I would like to make the following minor suggestions:
- The figures are helpful additions to the manuscript, but often have unlabelled or basic/default labels (eg “timefact” or “finalbin”) rather than appropriate descriptive labels
- In Fig 5B, the lowest right-hand panel has a straight line, which seems to be an error
- Fig 6A shows a linear increase in Shannon index over the study period, which does not seem accurate, and the figure does not add to the text
- It may be worthwhile stating that there would not be expected to be significant variance in results from samples obtained from inpatients (baseline data) vs. outpatients due to differences in handling or processing
- In the abstract, “corticosteroids” is plural and so “CS is” should read “CS are”.
